# Mathematical Model Describing HIV Infection with Time-Delayed CD4 T-Cell Activation

**Hernán Darío Toro-Zapata \***, **Carlos Andrés Trujillo-Salazar** and
**Edwin Mauricio Carranza-Mayorga**

Licenciatura en Matemáticas, Universidad del Quindío, Quindío 630004, Colombia;
catrujillo@uniquindio.edu.co (C.A.T.-S.); mao4021@hotmail.com (E.M.C.-M.)

\* Correspondence: hdtoro@uniquindio.edu.co

**Abstract:** A mathematical model composed of two non-linear differential equations that describe the population dynamics of CD4 T-cells in the human immune system, as well as viral HIV viral load, is proposed. The invariance region is determined, classical equilibrium stability analysis is performed by using the basic reproduction number, and numerical simulations are carried out to illustrate stability results. Thereafter, the model is modified with a delay term, describing the time required for CD4 T-cell immunological activation. This generates a two-dimensional integro-differential system, which is transformed into a system with three ordinary differential equations. For the new model, equilibriums are determined, their local stability is examined, and results are studied by way of numerical simulation.

**Keywords:** mathematical model; delay differential equations; HIV; immune system

---

## 1. Introduction

Acquired Immunodeficiency Syndrome (AIDS) represents a risk for a large part of the global population because it affects both people with the virus, as well as, indirectly, their families and friends. Historically, AIDS has been among the most-significant public health problems worldwide, owing to the number of people it has infected and the deaths it has caused. Additionally, it constitutes a social problem, as it has inhibited poverty reduction efforts. Since its emergence in the 1980s, AIDS has generated considerable challenges for the scientific community, which include understanding its dynamics and the search for a definitive cure [1–3].

Acquired Immunodeficiency Syndrome is a group of clinical symptoms produced as a result of the immune system's destruction caused by the Human Immunodeficiency Virus (HIV). The virus makes it impossible for the body to defend itself and facilitates the emergence of opportunistic illnesses. For this reason, AIDS has caused a considerable number of human deaths, primarily in regions with inadequate prevention mechanisms [3,4]. The Joint United Nations Programme on HIV/AIDS (UNAIDS) reported in 2018 the estimated number of people (all ages) living with HIV at 37.9 million [32.7–44.0] of which, 23.3 million people were receiving antiretroviral treatment by the end of 2018, equivalent to 62% of people living with HIV. However, it is cause for concern that 1.7 million [1.4–2.3] new cases are reported, implying a rate of 0.24 [0.18–0.31] new HIV infections (per 1000 uninfected population) [5].

To date, no cure has been found for HIV infection, which makes relevant the development of strategies that impede the virus' dispersion. Undoubtedly, the greatest risk of infection stems from sexual intercourse. However, other contagion factors exist, including blood transfusions, inadequate use of surgical materials (needles, scalpels, etc.), and vertical transmission (from mother to child) [6].

To briefly explain the dynamics of the HIV virus at cellular level, it must be mentioned that infection begins when the virus crosses the body's physical barriers and it is detected by dendritic cells (leucocytes that play an important role in the body's immunity) and macrophages, which are antigen-producing cells (antigens are substances that trigger antibody formation and may cause immune response). Once the virus is identified, CD4 T-cells come into play; they specialize in recognizing viral proteins and are tasked with activating humoral and cellular immune responses to fight infection. It is precisely the CD4 T-cells that are the virus' main target, which become infected and lose the ability to recognize vital proteins. This, together with the virus' high replication rate and its evasion strategies (use of reservoirs, sanctuaries, or dormant states), causes the body to gradually lose its ability to control the HIV infection [7].

A wide variety of mathematical models has been formulated to study HIV infection at cellular level [8–27], as well as its spread in uninfected populations [28–33]. Such models address the study of infection from different perspectives and have made it possible to evaluate the possible effect of preventive measures or prophylaxis and diagnosis on reducing transmission [29,31,34–37]. Concerning the effectiveness of antiretroviral medications in controlling viral loads, optimal control models have been very important tools [11,13,17,25,32,36–38]. Finally, any study must begin with knowledge of the virus and how it interacts with the host's immune system. Particularly, a more relevant topic is latent infection in people who are unaware of its serological status. One of the reasons for the slow development of the infection in people carrying the virus is the state of latency in which a proportion of the infected cells is found; indeed, this cell population consists of resting T cells that have not been immunologically activated, different from the activated T cells, also known as helper T-lymphocytes. The importance of this distinction is that HIV infects resting lymphocytes, but only 1% actively replicate viruses, which indicates that in 99% the pro-viral genome is housed and dormant [7–9,15–18,21,26,27,39].

Considering the above, and with the goal of studying CD4 T-cell interaction with viral HIV particles, two mathematical models, based on ordinary differential equations, are proposed here and will be studied from a qualitative perspective. First, the HIV infection dynamics at cellular level is addressed, without considering immunological activation, which implies that it is assumed that viral replication begins immediately after cell infection; then, these results are compared with those from a second model, incorporating a delay time that considers immunological activation as a determining factor in the dynamics of cell-to-cell HIV infection. Although the results obtained do not constitute a definitive solution to the problem, they do contribute theoretically to its comprehension.

The model stability analysis allows for concluding conditions on the parameters that would guarantee reduction of the viral load to undetectable levels, fundamentally reducing the value of the basic reproduction number $R_0$. This reduction can be achieved with adequate antiretroviral therapy strategies. Additionally, it is illustrated that under certain parameter values, the system may present Hopf bifurcation, which implies alternation, sustained over time, of high and low levels of viremia in the host, significantly affecting their quality of life. These cyclical scenarios should be avoided; in this sense, the results of this work study one of the parameters of the model that relates cyclical behavior to the CD4 T-cell activation time.

## 2. Model without CD4 T-Cell Activation Time

### 2.1. Model Formulation

Formulation of this model is based on predator-prey dynamics, supposing that increased viral load depends on the number of infected T cells that later die. This is a legitimate consideration due to the virus' high replication rate. The assumption is made that infected cells are instantly activated and produce viral particles, which are simplifications considered in studies including [13,20]. However, in Section 3, this assumption is weakened when time delay, which describes the time cells require to activate (a waiting time which implies a delay in the onset of viral replication) is included.

Let $T = T(t)$ be the average concentration of CD4 T-cells in a time $t$ and $V = V(t)$ be the average viral load in a time $t$. The constant rate of CD4 T-cell release is indicated by $\sigma$ and $\mu$ represents the rate of natural death. Additionally, the fact that proliferation of CD4 T-cells in response to the infection follows that the logistic law of growth is considered, where $\alpha$ is the proliferation rate (in general, it is $\alpha > \mu$ [40]) and $k$ is the carrying capacity. The effective contact rate between CD4 T-cells and virus is denoted by $\beta$, such that the $\beta TV$ (by the mass action law) represents the average concentration of infected T cells created in a given time $t$. Thus, the differential equation for the uninfected CD4 T-cell population is:

$$\dot{T} = \sigma + \alpha T \left( 1 - \frac{T}{k} \right) - \beta TV - \mu T.$$

A simplification considered in studies, such as [11,13,14] that consists of excluding an explicit state variable for the infected cell population, will be considered here. Thus, in each instant $t$, an average of $\beta TV$ cells become infected. The constant death rate of infected cells, owing to causes associated with viral replication, is indicated by $\delta$ and it is assumed that the number of infected cells that die in instant $t$ is proportional to those created. As such, the average concentration of infected cells that die in each instant $t$ is $\delta(\beta TV)$. That said, if each infected cell produces $N$ viral particles during its infectious period, the expression $N(\delta \beta TV)$ represents the average viral load produced in instant $t$. If $\eta = N\delta\beta$, then $\eta TV$ determines the number of free viral particles produced in a time $t$. This $\eta$ parameter is comparable to the biomass conversion rate (growth rate) seen in predator-prey models. Finally, it is assumed that viral particles are eliminated at proportionality rate $c$. The model takes the following form:

$$\begin{cases} \dot{T} = \sigma + \alpha T \left( 1 - \frac{T}{k} \right) - \beta TV - \mu T \\ \dot{V} = \eta TV - cV, \end{cases} \tag{1}$$

where the parameters $\sigma$, $\alpha$, $k$, $\beta$, $\mu$, $\eta$ and $c$ are positive. System (1) has non-negative initial conditions $T(0)$ and $V(0)$. Table 1 contains a description of state variables and model parameters.

**Table 1.** Description of state variables, initial conditions, and parameters used in simulations and the respective references.

| | State Variables | Initial Conditions | Reference |
|---|---|---|---|
| $T$ | Uninfected CD4 T-cell concentration | 500 mm$^{-3}$ | [13,19] |
| $V$ | Infectious viral load | 1 mm$^{-3}$ | [13,19] |
| | **Parameters** | **Value** | **Reference** |
| $\sigma$ | Constant CD4 T-cell production | 10 mm$^{-3}$d$^{-1}$ | [17,22,41] |
| $\alpha$ | Uninfected CD4 T-cell proliferation rate | 0.03 d$^{-1}$ | [19,40] |
| $\mu$ | Uninfected CD4 T-cell natural death rate | 0.01d$^{-1}$ | [17,22,41] |
| $\delta$ | Infected CD4 T-cell death rate | 0.26 d$^{-1}$ | [17,40] |
| $\beta$ | Effective contact rate between CD4 T-cells and virus | $6 \times 10^{-6}$ and $4 \times 10^{-5}$ mm$^3$d$^{-1}$ | [13,17,19,40] |
| $N$ | Number of viral particles produced per infected cell | 1000 | [41] |
| $c$ | Viral elimination rate | 2.4 d$^{-1}$ | [17,19] |
| $k$ | CD4 T-cell carrying capacity | 1500 mm$^{-3}$ | [19,40] |

Now that the model has been described, a proposition corresponding to the invariance region is enunciated and demonstrated; it corresponds to a set of biological interest, where the solutions are bounded and preserve their positivity over time.

**Proposition 1.** *If initial conditions $T(0)$ and $V(0)$ are non-negative, then the solutions to system (1) are limited to the positively invariant region given by:*

$$\Omega = \left\{ (T, V) \in \mathbb{R}^2 \mid 0 \le T \le k, 0 \le V \le \frac{M\eta}{c\beta} \right\}$$

with $M = \sigma + \frac{k\alpha}{4} + ck$.

**Proof.** This demonstration was carried out by considering procedures similar to those in [19,42], which suggest certain restrictions on the parameters to ensure that the model describes a realistic population dynamic. Particularly, it is assumed that $k > \frac{\sigma}{\mu}$. When this not the case, the population could grow beyond $k$. Considering this from the first equation in System (1),

$$
\begin{aligned}
\dot{T} &= \sigma - \mu T + \alpha T \left(1 - \frac{T}{k}\right) - \beta TV \\
&\le \sigma - \mu T + \alpha T \left(1 - \frac{T}{k}\right).
\end{aligned}
$$

As such, in $T = k$,

$$\dot{T} \le \sigma - \mu k < 0,$$

therefore, in the case of HIV infection, the T cell population is limited by $k$. Further, by defining function

$$\omega(t) = T(t) + \frac{\beta}{\eta} V(t),$$

and calculating the derivative of $\omega$ with respect to time along the trajectories of system (1), the following is obtained:

$$
\begin{aligned}
\dot{\omega} &= \dot{T} + \frac{\beta}{\eta} \dot{V} \\
&= \sigma + \alpha T \left(1 - \frac{T}{k}\right) - \left(\mu T + \frac{\beta}{\eta} cV\right) \\
&\le \sigma + \alpha T \left(1 - \frac{T}{k}\right).
\end{aligned}
$$

As the maximum value of the quadratic expression $\sigma + \alpha T \left(1 - \frac{T}{k}\right)$ is $\sigma + \frac{\alpha}{4}k$ when $0 < T \le k$, then,

$$
\begin{aligned}
\dot{\omega} + c\omega &= \sigma + \alpha T \left(1 - \frac{T}{k}\right) - \mu T - \frac{\beta}{\eta} cV + c \left(T + \frac{\beta}{\eta} V\right) \\
&= \sigma + \alpha T \left(1 - \frac{T}{k}\right) + cT - \mu T \\
&\le \sigma + \alpha T \left(1 - \frac{T}{k}\right) + cT \\
&\le \sigma + \frac{k\alpha}{4} + ck = M,
\end{aligned}
$$

therefore, when $t \to \infty$, $0 \le \omega \le \frac{M}{c}$ [43]. Thus,

$$
\begin{aligned}
\omega \le \frac{M}{c} \quad &\Leftrightarrow \quad T + \frac{\beta}{\eta} V \le \frac{M}{c} \\
&\Rightarrow \quad \frac{\beta}{\eta} V \le \frac{M}{c} \\
&\Rightarrow \quad V \le \frac{M\eta}{c\beta}.
\end{aligned}
$$

As such, $V \to \frac{M\eta}{c\beta}$, when $t \to \infty$. In other words, the solutions to system (1) are limited to the positively invariant region $\Omega$. $\quad\square$

## 2.2. Stability Analysis

### 2.2.1. Virus-Free Equilibrium

It is now sought to qualitatively study the system (1), that is, without the explicit knowledge of its solutions, determine the behavior of these solutions after a long period. To do so, it is necessary

to determine the equilibrium solutions or stationary solutions (which do not change over time), this requires solving the algebraic system,

$$\begin{cases} \sigma + \alpha T(1 - \frac{T}{k}) - \beta TV - \mu T = 0 \\ \eta TV - cV = 0, \end{cases} \tag{2}$$

whose trivial solutions are given by $E_0 = (\overline{T}_0, 0)$ and $E_0^- = (T_0^-, 0)$, which correspond to the absence of infection, where

$$\overline{T}_0 = \frac{k\mu}{2\alpha} \left[ (\xi_0 - 1) + \sqrt{(\xi_0 - 1)^2 + \frac{4\alpha\sigma}{k\mu^2}} \right]$$

and

$$T_0^- = \frac{k\mu}{2\alpha} \left[ (\xi_0 - 1) - \sqrt{(\xi_0 - 1)^2 + \frac{4\alpha\sigma}{k\mu^2}} \right]$$

with $\xi_0 = \frac{\alpha}{\mu}$ is the ratio between the uninfected CD4 T-cell proliferation rate and natural death rate.

So far, two virus-free equilibria have been found, but now it is necessary to establish conditions that guarantee the biological meaning or ecological coherence of this virus-free equilibria, done in Propositions 2 and 3, stated and demonstrated ahead.

**Proposition 2.** *equilibrium $E_0 = (\overline{T}_0, 0)$ makes biological sense, i.e., $\overline{T}_0 > 0$.*

**Proof.** Taking into account that all parameters considered are positive, the expression $4k\alpha\sigma > 0$ is valid. Then,

$$\begin{aligned} 4k\alpha\sigma > 0 \quad &\Leftrightarrow \quad (k\alpha - k\mu)^2 + 4k\alpha\sigma > (k\alpha - k\mu)^2 \\ &\Leftrightarrow \quad \sqrt{(k\alpha - k\mu)^2 + 4k\alpha\sigma} > k\mu - k\alpha \\ &\Leftrightarrow \quad k\alpha - k\mu + \sqrt{(k\alpha - k\mu)^2 + 4k\alpha\sigma} > 0 \\ &\Leftrightarrow \quad k\mu \left[ \frac{\alpha}{\mu} - 1 + \frac{\sqrt{k^2\mu^2 \left( \frac{\alpha}{\mu} - 1 \right)^2 + 4k\alpha\sigma}}{k\mu} \right] > 0 \\ &\Leftrightarrow \quad \frac{k\mu}{2\alpha} \left[ (\xi_0 - 1) + \sqrt{(\xi_0 - 1)^2 + \frac{4\alpha\sigma}{k\mu^2}} \right] > 0. \end{aligned}$$

As such, $\overline{T}_0 > 0$. $\square$

This result is important to the extent that it establishes that $E_0$ is an equilibrium with biological sense; therefore, considering that there is no cure for HIV infection, it could be associated with undetectable viral load levels.

**Proposition 3.** *Equilibrium $E_0^- = (T_0^-, 0)$ does not make biological sense, i.e., $T_0^- < 0$.*

**Proof.** Taking into account that all parameters considered are positive, $\frac{4\alpha\sigma}{k\mu^2} > 0$ is satisfied. Then,

$$\begin{aligned} \frac{4\alpha\sigma}{k\mu^2} > 0 \quad &\Leftrightarrow \quad (\xi_0 - 1)^2 + \frac{4\alpha\sigma}{k\mu^2} > (\xi_0 - 1)^2 \\ &\Leftrightarrow \quad \sqrt{(\xi_0 - 1)^2 + \frac{4\alpha\sigma}{k\mu^2}} > (\xi_0 - 1) \\ &\Leftrightarrow \quad 0 > (\xi_0 - 1) - \sqrt{(\xi_0 - 1)^2 + \frac{4\alpha\sigma}{k\mu^2}} \\ &\Leftrightarrow \quad 0 > \frac{k\mu}{2\alpha} \left[ (\xi_0 - 1) - \sqrt{(\xi_0 - 1)^2 + \frac{4\alpha\sigma}{k\mu^2}} \right]. \end{aligned}$$

As such, $T_0^- < 0$. $\square$

From the previous proposition, we discarded $E_0^-$ as a constant solution of the system and, therefore, it will not be discussed in the following.

It is important to investigate the conditions under which the infection enters the $HIV^+$ patient's immune system and potentially establishes in it, i.e., the conditions that promote continued infection. This information is obtained partially by the study of $E_0$ stability. In general, the basic reproduction number denoted by $R_0$ acts as a threshold, which determines the conditions with no risk of outbreak. Theoretically, if $R_0 < 1$ the virus is eliminated, but if $R_0 > 1$ an infection outbreak occurs. The $R_0$ is tied both to the strength of the means of viral infection transmission and the duration of epidemiological periods. As such, it is defined as the average concentration of secondary infections caused by a single infected CD4 T-cell in a population of entirely uninfected CD4 T-cells throughout its infectious period [12,23–25].

In order to determine $R_0$, consider that in an entirely uninfected T-cell population a small initial virus load ($V > 0$) is introduced. Conditions under which said virus concentration increases is obtained by imposing the condition $\dot{V} > 0$, then from the second equation in (1),

$$V\left(\eta T - c\right) > 0 \quad \Leftrightarrow \quad \eta T - c > 0$$
$$\Leftrightarrow \quad \frac{\eta T}{c} > 1.$$

Considering that $\overline{T}_0 = \frac{k\mu}{2\alpha}\left[(\xi_0 - 1) + \sqrt{(\xi_0 - 1)^2 + \frac{4\alpha\sigma}{k\mu^2}}\right]$ is the uninfected cell population in the absence of infection, and $\eta = N\beta\delta$, then $R_0$ is defined as,

$$R_0 = \frac{N\beta\delta}{c}\overline{T}_0. \tag{3}$$

To study the stability of the virus-free equilibrium $E_0$, we begin by a linearizing system (1) through the Jacobian matrix,

$$\mathbf{A}(T, V) = \begin{pmatrix} \alpha - \frac{2\alpha T}{k} - \beta V - \mu & -\beta T \\ \eta V & c\left(\frac{\eta}{c}T - 1\right) \end{pmatrix}. \tag{4}$$

The matrix (4) is evaluated at $E_0$, and the sign of the real part of its eigenvalues is studied, as shown in the following proposition's proof.

**Proposition 4.** *The virus-free equilibrium $E_0$ from system (1) is locally asymptotically stable if and only if $R_0 < 1$.*

**Proof.** Evaluation of the Jacobian matrix (4) in the virus-free equilibrium $E_0 = (\overline{T}_0, 0)$ results in the following:

$$\mathbf{A_0} = \mathbf{A}(E_0) = \begin{pmatrix} \alpha - \mu - \frac{2\alpha\overline{T}_0}{k} & -\beta\overline{T}_0 \\ 0 & c\left(\frac{\eta}{c}\overline{T}_0 - 1\right) \end{pmatrix}.$$

As $\mathbf{A_0}$ is a superior triangular matrix, its eigenvalues are $\lambda_1 = \alpha - \mu - \frac{2\alpha\overline{T}_0}{k}$ and $\lambda_2 = c\left(\frac{\eta}{c}\overline{T}_0 - 1\right)$. Firstly, $\lambda_1 < 0$ will be demonstrated, based on the equivalence of the following inequalities:

$$\frac{\sqrt{(k\alpha-k\mu)^2+4k\alpha\sigma}}{k\alpha} > 0 \quad \Leftrightarrow \quad \frac{\mu}{\alpha} - \frac{k\mu}{k\alpha} + \frac{\sqrt{(k\alpha-k\mu)^2+4k\alpha\sigma}}{k\alpha} + 1 > 1$$

$$\Leftrightarrow \quad \frac{\mu}{\alpha} + \frac{k\alpha-k\mu+\sqrt{(k\alpha-k\mu)^2+4k\alpha\sigma}}{k\alpha} > 1$$

$$\Leftrightarrow \quad \frac{\mu}{\alpha} + \frac{2}{k}\left(\frac{k\alpha-k\mu+\sqrt{(k\alpha-k\mu)^2+4k\alpha\sigma}}{2\alpha}\right) > 1$$

$$\Leftrightarrow \quad \frac{\mu}{\alpha} + \frac{2\overline{T}_0}{k} > 1$$

$$\Leftrightarrow \quad \mu + \frac{2\alpha\overline{T}_0}{k} > \alpha.$$

Thus, it is revealed that $\alpha - \mu - \frac{2\alpha\overline{T}_0}{k} < 0$ and so, $\lambda_1 < 0$. On the other hand, it is evident that $\lambda_2 = c\left(\frac{\eta}{c}\overline{T}_0 - 1\right) = c\left(R_0 - 1\right) < 0$, if and only if $R_0 < 1$. Given that both eigenvalues are negative, it is concluded that the virus-free equilibrium $E_0$ is locally asymptotically stable. $\quad\square$

The result just demonstrated refers to the hypothetical situation under which the virus is completely removed from the immune system. However, it is known that HIV infection, even today, does not have a cure that allows reaching a stage of definitive viral elimination. In this sense, from a clinical point of view, condition $R_0 < 1$ should be interpreted as that scenario in which a patient carrying the virus reaches an undetectable viral load.

### 2.2.2. Endemic Equilibrium

In this subsection, we determined the endemic equilibrium at which the infection is present, i.e., a constant solution with $V > 0$. From system (2), $T$ is cleared from the second equation and the equilibrium coordinate is,

$$\overline{T}_1 = \frac{c}{\eta},$$

which is replaced into the first equation of system (2) to obtain

$$\sigma + \alpha\frac{c}{\eta}\left(1 - \frac{c}{\eta k}\right) - \beta\frac{c}{\eta}V - \mu\frac{c}{\eta} = 0,$$

then, the $V$ coordinate at the equilibrium is given by

$$\begin{aligned}\overline{V}_1 &= \frac{1}{\beta}\left(\frac{\eta}{c}\sigma + \alpha\left(1 - \frac{c}{k\eta}\right) - \mu\right) \\ &= \frac{1}{\beta}\left(\frac{k\eta^2\sigma+ck\alpha\eta-(c^2\alpha+ck\eta\mu)}{ck\eta}\right) \\ &= \frac{(c\alpha+k\eta\mu)(\xi_1-1)}{k\beta\eta},\end{aligned}$$

where

$$\xi_1 = \frac{k\eta(c\alpha + \eta\sigma)}{c(c\alpha + k\eta\mu)}, \tag{5}$$

therefore, the endemic $E_1 = (\overline{T}_1, \overline{V}_1)$ equilibrium is given by

$$E_1 = \left(\frac{c}{\eta}, \frac{(c\alpha + k\eta\mu)(\xi_1 - 1)}{k\beta\eta}\right). \tag{6}$$

Note that equilibrium $E_1$ makes biological sense when $\xi_1 > 1$, case in which $\overline{V}_1 > 0$. The intention now is to study the stability of the endemic equilibrium $E_1$, which is done in Proposition 5 from the basic reproduction number $R_0$, but whose demonstration requires the following Lemma:

**Lemma 1.** $R_0 > 1$ *if and only if* $\xi_1 > 1$.

**Proof.** Indeed,

$$R_0 > 1 \quad \Leftrightarrow \quad \frac{\eta}{c}\frac{k\mu}{2\alpha}\left[(\zeta_0 - 1) + \sqrt{(\zeta_0 - 1)^2 + \frac{4\alpha\sigma}{k\mu^2}}\right] > 1$$

$$\Leftrightarrow \quad \sqrt{(\zeta_0 - 1)^2 + \frac{4\alpha\sigma}{k\mu^2}} > \frac{2c\alpha}{k\eta\mu} - (\zeta_0 - 1)$$

$$\Leftrightarrow \quad (\zeta_0 - 1)^2 + \frac{4\alpha\sigma}{k\mu^2} > \left[\frac{2c\alpha}{k\eta\mu} - (\zeta_0 - 1)\right]^2$$

$$\Leftrightarrow \quad \frac{4\alpha\sigma}{k\mu^2} > \frac{4c^2\alpha^2}{k^2\eta^2\mu^2} - \frac{4c\alpha}{k\eta\mu}(\zeta_0 - 1).$$

When both sides are multiplied by $\frac{k\mu^2\eta}{4\alpha}$ and considering that $\zeta_0 = \frac{\alpha}{\mu}$,

$$R_0 > 1 \quad \Leftrightarrow \quad \sigma\eta > \frac{\alpha c^2}{k\eta} - c\mu\left(\frac{\alpha}{\mu} - 1\right)$$

$$\Leftrightarrow \quad \sigma\eta - \frac{\alpha c^2}{k\eta} + c\alpha - c\mu > 0$$

$$\Leftrightarrow \quad \sigma\eta k\eta - \alpha c^2 + c\alpha k\eta - c\mu k\eta > 0$$

$$\Leftrightarrow \quad k\eta(\sigma\eta + c\alpha) - c(c\alpha + \mu k\eta) > 0$$

$$\Leftrightarrow \quad c(\alpha c + \mu k\eta)\left(\frac{k\eta(\sigma\eta + c\alpha)}{c(\alpha c + \mu k\eta)} - 1\right) > 0$$

$$\Leftrightarrow \quad \frac{k\eta(\sigma\eta + c\alpha)}{c(\alpha c + \mu k\eta)} > 1$$

$$\Leftrightarrow \quad \zeta_1 > 1.$$

□

**Proposition 5.** *The endemic equilibrium $E_1$ from system (1) makes biological sense and it is locally asymptotically stable, if and only if $R_0 > 1$.*

**Proof.** By evaluating the Jacobian matrix (4) in the endemic equilibrium $E_1$ as described in (6) results in

$$\mathbf{A_1} = \mathbf{A}(E_1) = \begin{pmatrix} \alpha - \mu - \frac{2\alpha\overline{T}_1}{k} - \beta\overline{V}_1 & -\beta\overline{T}_1 \\ \eta\overline{V}_1 & 0 \end{pmatrix}.$$

From which the trace and determinant of $\mathbf{A_1}$ are, respectively,

$$\text{tr}(\mathbf{A_1}) = \alpha - \mu - \frac{2\alpha\overline{T}_1}{k} - \beta\overline{V}_1$$

$$\det(\mathbf{A_1}) = \beta\eta\overline{T}_1\overline{V}_1.$$

Firstly, $\text{tr}(\mathbf{A_1}) < 0$, a result obtained by replacing $\overline{T}_1$ and $\overline{V}_1$ and simplifying, as shown:

$$\text{tr}(\mathbf{A_1}) = -\left(\frac{\alpha c^2 + \eta^2\sigma k}{\eta ck}\right) < 0.$$

Secondly, it is clear that $\overline{T}_1 = \frac{c}{\eta} > 0$ and according to (6) and Lemma 1, it is verified that $\overline{V}_1 > 0$ if and only if $R_0 > 1$, thus $\det(\mathbf{A_1}) > 0$. Then, by the trace-determinant criteria, the proposition is proven. □

*2.3. Simulation without CD4 T-Cell Activation Time*

For the model without CD4 T-cell activation time described in (1), two numerical simulations are performed to graphically illustrate the stability results obtained analytically. These two scenarios are associated with the basic reproduction number $R_0$, specifically $R_0 < 1$ in Figure 1 and $R_0 > 1$ in Figure 2. In both cases, the solution curves associated with the CD4 T-cell populations $T$ and viral load $V$ are shown in the top panels and the corresponding $(T, V)$–phase space in the bottom panel. The numerical value of $R_0$ is shown to notice the incidence it has on the system's behavior. The initial

conditions considered for the simulation are $T(0) = 500$ and $V(0) = 1$ and the parameter values are shown in Table 1, while different values for the effective contact rate between CD4 T-cells and the virus $\beta$ are assigned.

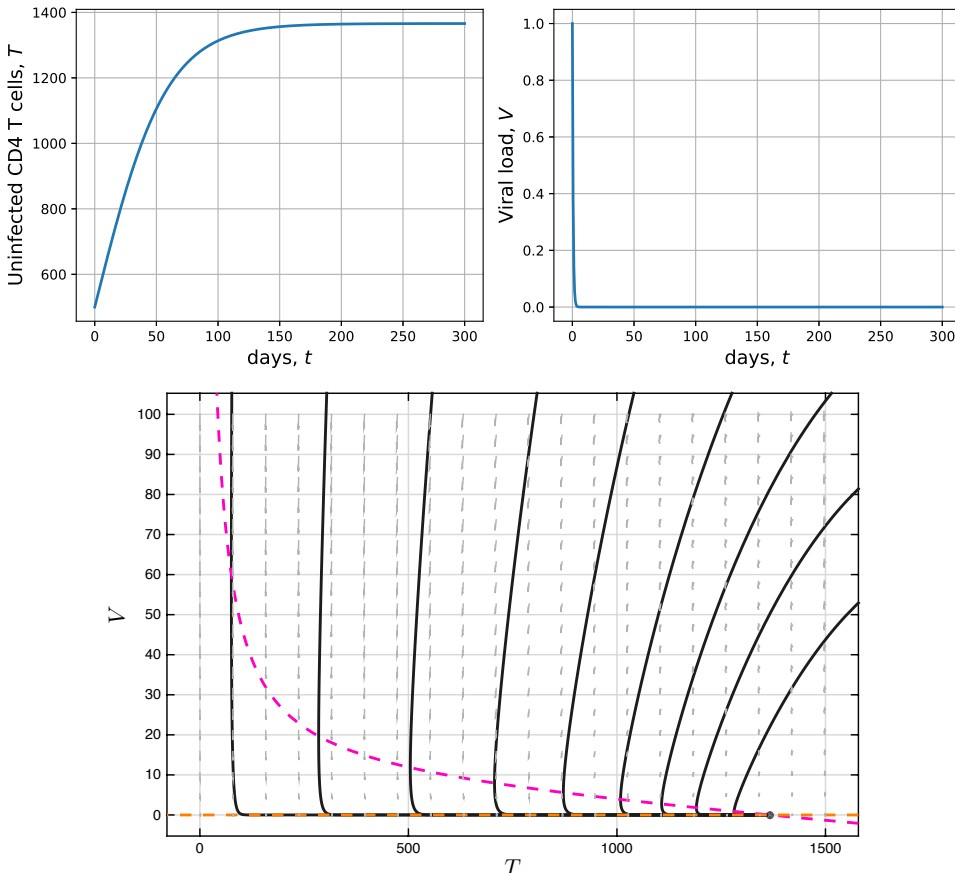

**Figure 1.** Numeric solutions vs. time (**top panels**) and phase space (**bottom panel**), corresponding to uninfected CD4 T-cells and viral load in model (1), with $\beta = 6 \times 10^{-6}$ and $R_0 = 0.8953$ are shown. In this scenario the virus-free equilibrium $E_0$ is locally asymptotically stable.

For the first simulation in Figure 1, $\beta = 6 \times 10^{-6}$ was used to have $R_0 = 0.8953$. Given that $R_0 < 1$, the virus-free equilibrium $E_0 = (1366, 0)$ is locally asymptotically stable. Note that CD4 T-cells achieve equilibrium after 150 days, approximately, while the viral load is immediately eliminated. The phase portrait in the bottom panel shows, in dashed magenta and dashed yellow, the null-clines $\dot{T} = 0$ and $\dot{V} = 0$, respectively; therefore, the black intersection point at $(1366, 0)$ corresponds to the virus-free equilibrium $E_0$. Several numerical solutions are shown and depicted by means of solid black lines that describe the state trajectories toward $E_0$. In this scenario, the virus-free equilibrium presents a stable node behavior [44]. As discussed above in the virus-free equilibrium stability analysis, this scenario, while desirable, cannot be definitively achieved, i.e., the virus cannot be eliminated from the organism of an HIV$^+$ patient. However, it is clear that antiretroviral treatment therapies must be focused on reducing the value of $R_0$ to bring the HIV$^+$ patient to undetectable viral load levels.

In Figure 2 the simulation for $R_0 > 1$ is shown. The values mentioned in Table 1 were considered again, but the value of the effective contact rate between CD4 T-cells and the virus was increased, specifically $\beta = 4 \times 10^{-5}$, to have $R_0 = 5.0175$. As $R_0 > 1$, the endemic equilibrium $E_1 = (231, 1468)$, is locally asymptotically stable. Indeed, in both, the CD4 T-cell concentration $T$ and viral load $V$, oscillations were present until achieving the corresponding equilibriums, as illustrated in the top panels. It is noteworthy that for uninfected CD4 T-cells (left), the value $231/\text{mm}^3$ is reached, corresponding to a very low T-cell concentration and leading to the patient's need to initiate antiretroviral therapy.

Furthermore, it is important to underscore that viral concentration grows extremely quickly from the initial condition point $V(0) = 1$ until achieving a maximum of approximately 30,000 viral load per mm$^3$. From there, a set of damped oscillations follows until reaching equilibrium. In the bottom panel, the phase portrait shows the stable spiral-type behavior of the solution (solid black line) [44]. As in the previous case, in dashed magenta and dashed yellow, the null-clines $\dot{T} = 0$ and $\dot{V} = 0$, respectively, are shown; therefore, the intersection points between those lines correspond to the system's equilibria.

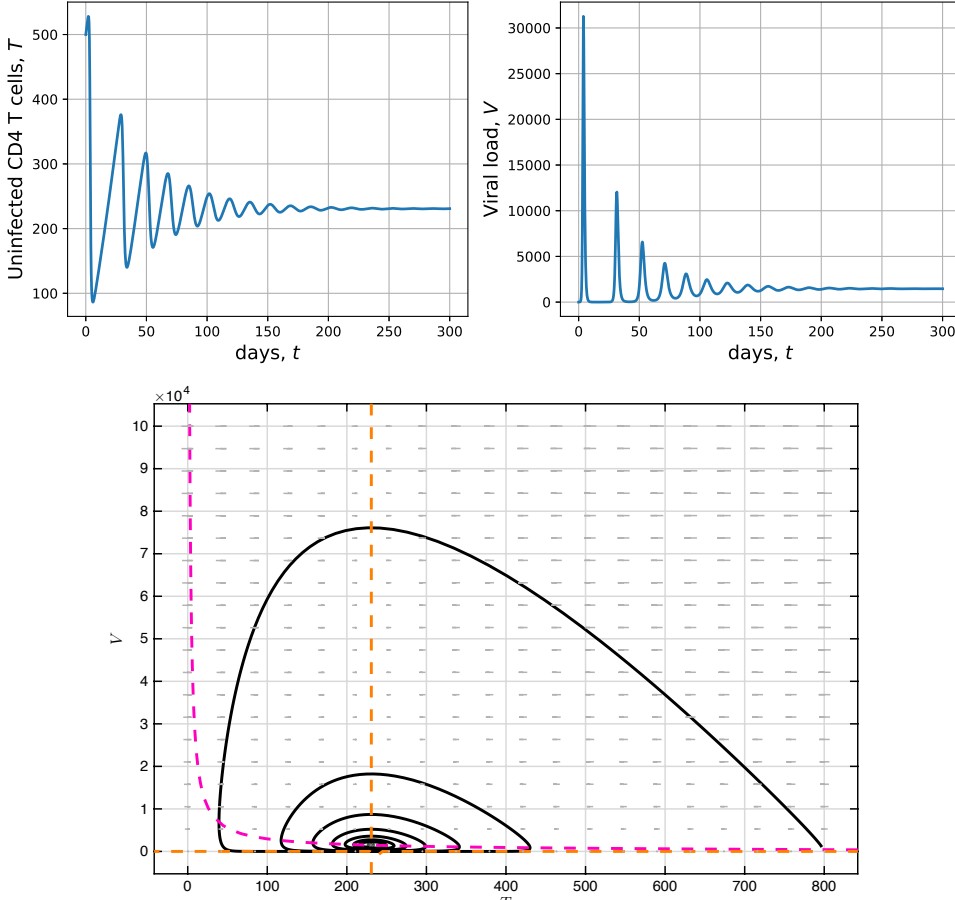

**Figure 2.** Numeric solutions vs. time (**top panels**) and phase space (**bottom panel**), corresponding to uninfected CD4 T-cells and viral load in model (1), with $\beta = 4 \times 10^{-5}$ and $R_0 = 5.0175$ are shown. In this scenario, the endemic equilibrium $E_1$ is locally asymptotically stable.

## 3. Model with CD4 T-Cell Activation Time

### 3.1. Model Formulation

The uninfected CD4 T-cell population consists of resting T cells that have not been immunologically activated and of activated T cells, also known as helper T-lymphocytes. The importance of this distinction is that HIV infects resting lymphocytes, but only 1% actively replicate, which indicates that in 99% the pro-viral genome is housed and dormant [7,16,17,21,39].

Assume, then, that $\tau$ is the average time that immunological activation of resting T cells requires converting into active HIV replicators. In system (1), it was assumed that $\eta T V$ was the average concentration of viral particles produced in a time $t$, with $\eta = N\delta\beta$. Consider that the number of new virions produced depends on the $T$ cells previously infected, which permits considering those infected during their idle state. One way to enact this consideration in a differential equation system is by incorporating a continuous delay term, as proposed by Volterra, and given by

$$\eta V \int_{-\infty}^{t} F(t - \tau)T(\tau)d\tau,$$

where $F$ is a factor considering the emphasis that should be given to the $T$ cell population size in the recent past. According to this, system (1) adopts the integro-differential form shown below:

$$\begin{cases} \dot{T} = \sigma + \alpha T \left( 1 - \frac{T}{k} \right) - \beta TV - \mu T \\ \dot{V} = \eta V \int_{-\infty}^{t} F(t - \tau)T(\tau)d\tau - cV. \end{cases} \tag{7}$$

Function $F$ in (7) is called *kernel* and has been used by authors, including [45–47], in the context of predator-prey models, but with different forms. On the topic, Ref. [48] states that $F$'s general form is

$$F_m(t) = \frac{a^m}{(m - 1)!} t^{m-1} e^{-at}, \tag{8}$$

with $a > 0$ and $m \in \mathbb{N}$. In [49], it is warned that cases different from $m = 1$ and $m = 2$ are minimally objective from the point of view of expected results, as from $m = 3$ significant changes in function behavior are not observed. Further, when $m$ tends toward infinity, the $F_m$ function tends toward the $\delta$ function. In our case, $m = 1$ is considered, then (8) takes the form,

$$F(t) = ae^{-at}; \quad a > 0.$$

Therefore, system (7) can be written as,

$$\begin{cases} \dot{T} = \sigma + \alpha T \left( 1 - \frac{T}{k} \right) - \beta TV - \mu T \\ \dot{V} = \eta V \int_{-\infty}^{t} ae^{-a(t-\tau)} T(\tau)d\tau - cV. \end{cases} \tag{9}$$

One way to study the integro-differential system (9) is to transform it into an ordinary differential equation system via the introduction of a new variable, suggested by [46] and defined by,

$$W(t) = \int_{-\infty}^{t} ae^{-a(t-\tau)} T(\tau)d\tau; \quad t \geq 0.$$

Expressing again, in equivalent form, the following results:

$$W = a \left[ \int_{-\infty}^{s} e^{-a(t-\tau)} T(\tau)d\tau + \int_{s}^{t} e^{-a(t-\tau)} T(\tau)d\tau \right],$$

with $s \leq t$. Deriving $W$ with respect to $t$:

$$\begin{aligned} \dot{W} &= a \left[ \int_{-\infty}^{s} \frac{\partial}{\partial t} \left[ e^{-a(t-\tau)} T(\tau) \right] d\tau + \frac{d}{dt} \int_{s}^{t} e^{-a(t-\tau)} T(\tau)d\tau \right] \\ &= a \left[ - \int_{-\infty}^{s} ae^{-a(t-\tau)} T(\tau)d\tau + T(t) \right] \\ &= a \left[ T(t) - \int_{-\infty}^{s} ae^{-a(t-\tau)} T(\tau)d\tau \right] \\ &= a(T - W). \end{aligned}$$

As such:

$$\dot{W} = a(T - W).$$

Thus, the integro-differential system (9) is transformed into a system of ordinary differential equations, which will be the object of study in the following sections and which is shown ahead,

$$\begin{cases} \dot{T} = \sigma + \alpha T \left(1 - \frac{T}{k}\right) - \beta TV - \mu T \\ \dot{V} = \eta VW - cV \\ \dot{W} = a(T - W). \end{cases} \tag{10}$$

The theory on the topological equivalence of systems (9) and (10) may be consulted in [46].

### 3.2. Stability Analysis with CD4 T-Cell Activation Time

In order to find System (10)'s stationary solutions, the algebraic system that results from equaling the derivatives to zero is solved and two equilibriums are obtained: trivial and endemic. The virus-free equilibrium is denoted by $E_0$, corresponding to the absence of viral load and given by $E_0 = (\overline{T}_0, \overline{V}_0, \overline{W}_0)$, where:

$$\begin{aligned} \overline{T}_0 &= \frac{k\mu}{2\alpha} \left[ (\xi_0 - 1) + \sqrt{(\xi_0 - 1)^2 + \frac{4\alpha\sigma}{k\mu^2}} \right] \\ \overline{V}_0 &= 0 \\ \overline{W}_0 &= \frac{k\mu}{2\alpha} \left[ (\xi_0 - 1) + \sqrt{(\xi_0 - 1)^2 + \frac{4\alpha\sigma}{k\mu^2}} \right]. \end{aligned}$$

This equilibrium makes biological sense because $\overline{T}_0 = \overline{W}_0$ and the positivity of $\overline{T}_0$ was previously studied in Proposition 2. Further, $E_0$ may be expressed in terms of $R_0$, as follows:

$$E_0 = (\overline{T}_0, \overline{V}_0, \overline{W}_0) = \left( \frac{c}{\eta} R_0, 0, \frac{c}{\eta} R_0 \right).$$

Moreover, endemic equilibrium $E_1 = (\overline{T}_1, \overline{V}_1, \overline{W}_1)$, where a non-zero viral load exists, presents the following coordinates:

$$\overline{T}_1 = \frac{c}{\eta}, \quad \overline{V}_1 = \frac{c\alpha + k\eta\mu}{k\beta\eta} (\xi_1 - 1) \quad \text{and} \quad \overline{W}_1 = \frac{c}{\eta}.$$

It is observed that the first two coordinates $\overline{T}_1$ and $\overline{V}_1$ coincide with those shown in (6), which correspond to the endemic equilibrium of system (1), whose biological sense was previously discussed with the help of Lemma 1. As $\overline{W}_1 = \frac{c}{\eta} > 0$, it is concluded that $E_1 = (\overline{T}_1, \overline{V}_1, \overline{W}_1)$ also makes biological sense.

We now analyze the stability of the virus-free equilibrium $E_0$ and endemic equilibrium $E_1$ in Propositions 6 and 7, respectively. In both cases, the Jacobian matrix given in (11) is required,

$$\mathbf{J}(T, V, W) = \begin{pmatrix} \alpha - \frac{2\alpha T}{k} - \beta V - \mu & -\beta T & 0 \\ 0 & \eta W - c & \eta V \\ a & 0 & -a \end{pmatrix}. \tag{11}$$

**Proposition 6.** *The virus-free equilibrium $E_0$ in system (10) is locally asymptotically stable, if and only if $R_0 < 1$.*

**Proof.** Evaluating the Jacobian matrix (11) in the virus-free equilibrium $E_0$ results in

$$\mathbf{J}(E_0) = \begin{pmatrix} \alpha - \mu - \frac{2\alpha \overline{T}_0}{k} & -\beta \overline{T}_0 & 0 \\ 0 & \eta \overline{W}_0 - c & 0 \\ a & 0 & -a \end{pmatrix},$$

whose characteristic equation is given by

$$(\lambda + a)(\lambda - \eta \overline{W}_0 + c)(k\lambda - k\alpha + k\mu + 2\alpha \overline{T}_0) = 0,$$

then, the corresponding eigenvalues are

$$\begin{aligned} \lambda_1 &= -a \\ \lambda_2 &= \eta \overline{W}_0 - c = \eta \overline{T}_0 - c = c\left(\frac{\eta}{c}\overline{T}_0 - 1\right) = c(R_0 - 1) \\ \lambda_3 &= \alpha - \mu - \frac{2\alpha \overline{T}_0}{k}. \end{aligned}$$

Clearly, $\lambda_1 < 0$. For $\lambda_2$ one has $\lambda_2 < 0$, if and only if $R_0 < 1$ and for $\lambda_3 < 0$ it must be ensured that $\alpha - \mu - \frac{2\alpha \overline{T}_0}{k} < 0$, which was previously verified in the proof of Proposition 4. Thus, it is concluded that virus-free equilibrium $E_0$ is locally asymptotically stable, if and only if $R_0 < 1$. $\square$

**Proposition 7.** *If $R_0 > 1$ and $a > a_0 = \frac{c^2(c\alpha + k\eta\mu)(\xi_1 - 1)}{c^2\alpha + k\eta^2\sigma}$, endemic equilibrium $E_1$ in system (10) is locally asymptotically stable.*

**Proof.** Evaluating the Jacobian matrix (11) in endemic equilibrium $E_1$ results in

$$\mathbf{J}(E_1) = \begin{pmatrix} \alpha - \frac{2\alpha \overline{T}_1}{k} - \beta \overline{V}_1 - \mu & -\beta \overline{T}_1 & 0 \\ 0 & 0 & \eta \overline{V}_1 \\ a & 0 & -a \end{pmatrix}$$

whose characteristic equation is given by:

$$\lambda^3 + b_1 \lambda^2 + b_2 \lambda + b_3 = 0$$

where coefficients $b_1$, $b_2$, and $b_3$ correspond to:

$$\begin{aligned} b_1 &= a + \mu - \alpha + \frac{2c\alpha + (c\alpha + k\eta\mu)(\xi_1 - 1)}{k\eta} \\ b_2 &= a\left[\mu - \alpha + \frac{2c\alpha + (c\alpha + k\eta\mu)(\xi_1 - 1)}{k\eta}\right] \\ b_3 &= \frac{ac(c\alpha + k\eta\mu)(\xi_1 - 1)}{k\eta} \end{aligned}$$

and where it is observed that:

$$b_1 = a + \frac{1}{a}b_2. \tag{12}$$

If in $b_1$ and $b_2$, $\xi_1$ is replaced as in (5), the following is obtained:

$$b_1 = \frac{c^2\alpha + ack\eta + k\eta^2\sigma}{ck\eta} > 0$$

$$b_2 = \frac{a\left(c^2\alpha + k\eta^2\sigma\right)}{ck\eta} > 0.$$

In addition, it is observed that $\xi_1 > 1$ implies $b_3 > 0$. As in Lemma 1, it was demonstrated that $R_0 > 1$, if and only if $\xi_1 > 1$, and so it is concluded that $R_0 > 1$ implies $b_3 > 0$. Conversely,

$$a > \frac{c^2(c\alpha + k\eta\mu)(\xi_1 - 1)}{c^2\alpha + k\eta^2\sigma} \quad \Leftrightarrow \quad a(c^2\alpha + k\eta^2\sigma) > c^2(c\alpha + k\eta\mu)(\xi_1 - 1)$$

$$\Leftrightarrow \quad a\left[\frac{a(c^2\alpha + k\eta^2\sigma)}{ck\eta}\right] > \frac{ac(c\alpha + k\eta\mu)(\xi_1 - 1)}{k\eta}$$

$$\Leftrightarrow \quad ab_2 > b_3$$

$$\Rightarrow \quad ab_2 + \frac{1}{a}b_2^2 > b_3$$

$$\Leftrightarrow \quad \left(a + \frac{1}{a}b_2\right)b_2 > b_3.$$

If on the left side of the last inequality, expression (12) is replaced, $b_1 b_2 > b_3$ is obtained. Then, based on the Routh-Hurwitz criteria [50], it is concluded that the equilibrium $E_1$ is locally asymptotically stable whenever $R_0 > 1$ and $a > a_0 = \frac{c^2(c\alpha + k\eta\mu)(\xi_1 - 1)}{c^2\alpha + k\eta^2\sigma}$. $\square$

### 3.3. Simulation with CD4 T-Cell Activation Time

To graphically illustrate the stability results obtained analytically for the model for CD4 T-cell with activation time described in (10), two numerical simulations were performed, again using the parameter values from Table 1. The state $W$ does not appear explicitly in the simulation, but it requires an initial condition, for which $W(0) = 0$ was considered. In this case, two scenarios associated with the threshold value $a$ from Proposition 7 are shown, specifically $a = 5$ and $a = 2$, displaying the numerical solutions associated with CD4 T-cell populations and viral load, in addition to phase-space depictions.

From the parameter values in Table 1, a numerical value for the threshold $a_0$ from Proposition 7 was obtained,

$$a_0 = \frac{c^2(c\alpha + k\eta\mu)(\xi_1 - 1)}{c^2\alpha + k\eta^2\sigma} = 2.9390374331550797,$$

therefore, in Figure 3, we set $a = 5 > a_0$, $T(0) = 500$ and $V(0) = 1$ are used as initial conditions; additionally, $\beta = 4 \times 10^{-5}$ was considered as in Figure 2 to compare both dynamics without and with CD4 T-cell activation times. As stated before, considering the parameters in Table 1, we get $R_0 = 5.0175$; therefore, the endemic equilibrium $E_1$ is locally asymptotically stable for both models. A numerical solution with CD4 T-cell activation time (delayed model (10)) is shown in solid orange, while a numerical solution without CD4 T-cell activation time (model (1)) is shown in dashed blue. As expected, both solutions tend towards the endemic equilibrium $E_1$. This coincides with the stability results from Propositions 4 and 7. It is important to note that the model considering CD4 T-cell activation time implies wider oscillations in both concentrations, which also implies that more time is needed to reach equilibrium state (a longer transient). It can also be observed that solutions with activation time are slightly delayed with respect to the solutions not considering activation time. Certainly, the magnitude of this delay is controlled by the value of parameter $a$. In particular, regarding the viral load concentration, it can be observed that it grows very quickly from the initial condition $V(0) = 1$, but reaches an approximate maximum average concentration >45,000 cells per mm$^3$; whereas, in the scenario without activation time that maximum was slightly >30,000 cells per mm$^3$ (see Figure 2, right top panel). In the bottom panel, the phase portrait is shown for $T$, $V$ and $W$ from the model

with CD4 T-cell activation time, in which the blue trajectory tends toward the endemic equilibrium, corresponding to an attractor spiral.

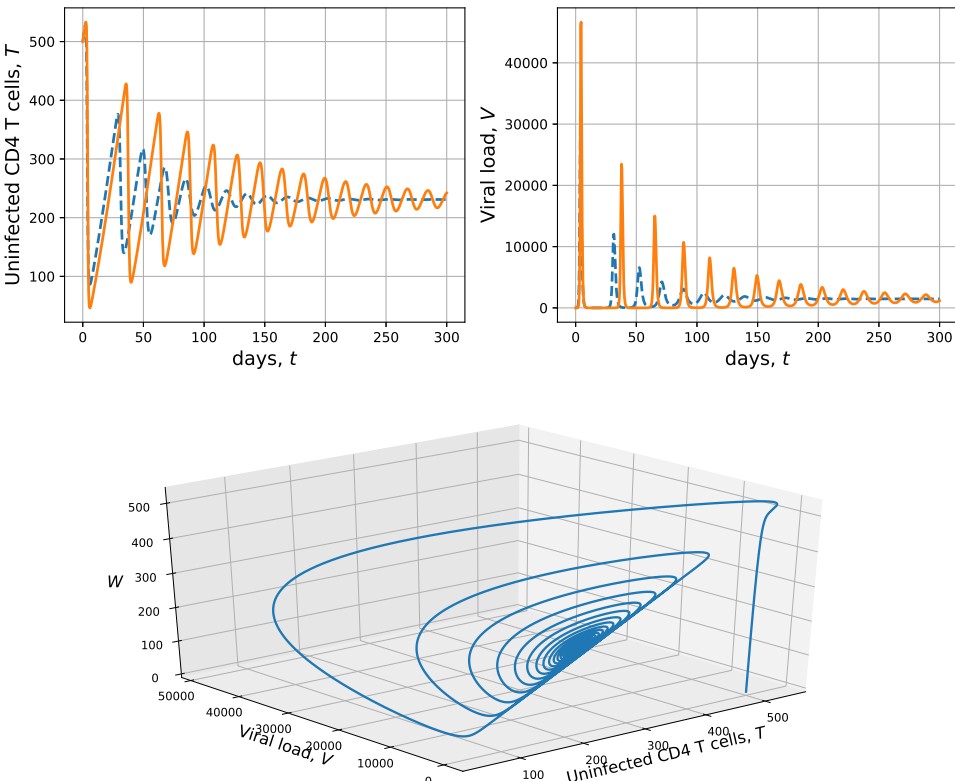

**Figure 3.** Numeric solutions vs. time (**top panels**) and phase space (**bottom panel**), corresponding to uninfected CD4 T-cells and viral load for the model without CD4 T-cell activation time (1) in dashed blue, and with CD4 T-cell activation time (10) in solid orange are shown. Both solutions are obtained for $\beta = 4 \times 10^{-5}$ to get $R_0 = 5.0175$, the parameters in Table 1 and $a = 5$ ($a > a_0$, as stated in Proposition 7). The initial conditions are $T(0) = 500$ and $V(0) = 1$. In this scenario, the endemic equilibrium $E_1$ is locally asymptotically stable.

For the simulation in Figure 4, we use $a = 2 < a_0$, case in which Proposition 7 does not decide on the stability of $E_1$. As in the previous cases, $\beta = 4 \times 10^{-5}$ and $R_0 = 5.0175 > 1$. The numerical results from model (10) are shown in solid blue. No equilibria stability is observed, indeed, the main characteristic related to this simulation is the periodic behavior. In the bottom panel, two solutions in the phase space are shown for $T(0) = 500$, $V(0) = 1$ and $W(0) = 0$ in solid blue and $T(0) = 230$, $V(0) = 1467$ and $W(0) = 0$ in solid orange. These solutions are not attracted by an equilibrium, but by a limit cycle surrounding the endemic equilibrium $E_1$. This situation reflects a critical state of the patient's immune system due to the alternation between moments of high immunodeficiency (which would require a strong intervention with antiretroviral treatments) and moments of relative immune strength.

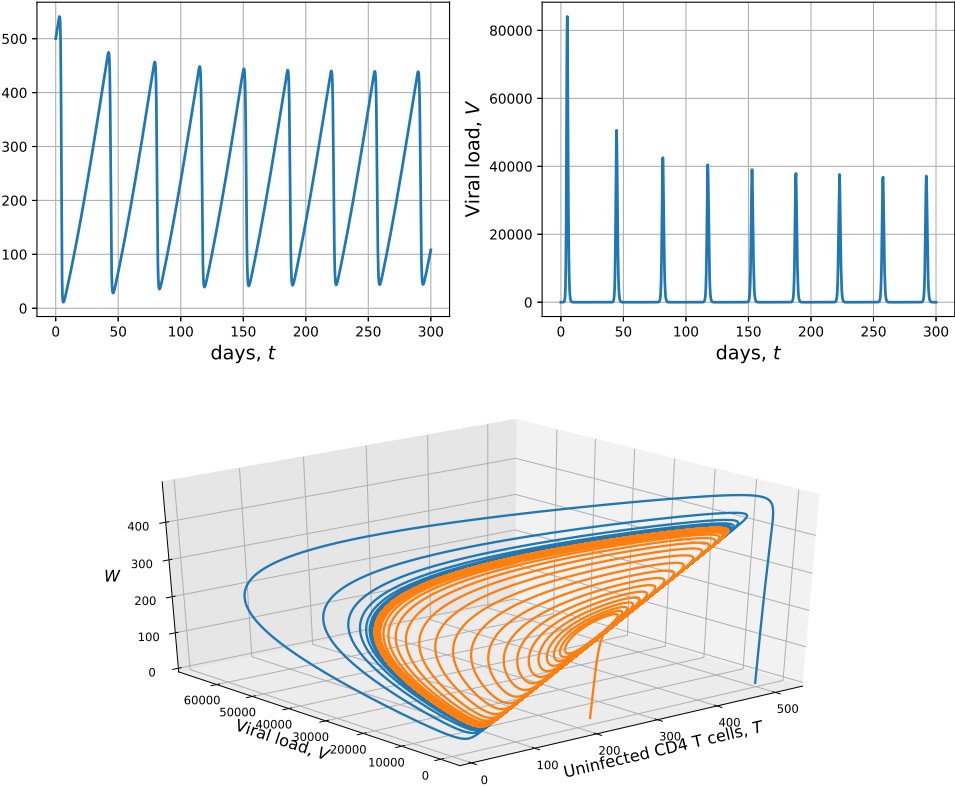

**Figure 4.** Numeric solutions vs. time (**top panels**) and phase space (**bottom panel**), which correspond to uninfected CD4 T-cells $T$ and viral load $(V)$ from Model (10), with $a = 2$, $\beta = 4 \times 10^{-5}$ and $R_0 = 5.0175$. In this scenario, the virus-free equilibrium $E_1$ is unstable. In the phase space, the initial conditions for the blue curve are $T(0) = 500$ and $V(0) = 1$, and the initial conditions for the orange curve are $T(0) = 230$ and $V(0) = 1467$.

## 4. Conclusions

The study of the model without CD4 T-cell activation time allowed knowing the local stability conditions of the system depending on the value of the basic reproduction number. Certainly, it is enough to bring that value <1 for the viral load of the HIV$^+$ patient to be controlled. This viral load control may be done by using antiretroviral therapy and it should focus on reducing the value of the model's own parameters, such as the rate of effective contact between CD4 T-cells and the virus, or the number of viral particles produced per infected cell.

The study of models for infection with HIV, with and without activation time, contributes to understanding the effect that this activation time has on the T cells and viral load dynamics. In general, considering that activation time delays the time of successive infection peaks, and wider oscillations are observed if compared with a scenario without activation time, which indicates a more profound lack of stability on the HIV$^+$ patient's immunological system. It is important to highlight that the threshold value $a_0$, given in Proposition 7, corresponds to a bifurcation parameter. Indeed, for $R_0 > 1$ and $a > a_0$, the endemic equilibrium is locally asymptotically stable, i.e., sustained viremia levels are reached by the host. However, for $R_0 > 1$ and $a < a_0$, a more in-depth analytical study can be performed; however, it was shown numerically that the endemic equilibrium was unstable. Indeed, according to [51], if the kernel in model (7) has the form (8), a Hopf bifurcation happens. This is verified, at least numerically, with the phase portrait shown in Figure 4, where a limit cycle is observed.

Mathematical models permit implementing different theories, which help to interpret and analyze an epidemiological problem, such as HIV infection. Our work presents a simple model, with only two state variables, that allows reflecting on the importance of considering intracellular activation times when modeling HIV infection. Although the study of HIV dynamics using simple models (like the one

presented here) is desirable, most robust models, involving the most crucial aspects of cell biology, can provide better predictions about the infection's dynamics. Finally, it should be considered that this type of model is quite sensitive to the parameter values, and thus, its use by decision makers should always be supported by clinical evidence and expert criteria.

**Author Contributions:** All authors have had a part in conceptualization, methodology, software, formal analysis, investigation, and writing-original draft preparation. All authors have read and agreed to the published version of the manuscript.

**Funding:** This research was funded by Universidad del Quindío, Armenia–Colombia, grant number 614.

**Acknowledgments:** The authors express sincere gratitude to the reviewers of the original manuscript. Their contributions have been very valuable and have significantly improved the paper's quality.

**Conflicts of Interest:** The authors declare no conflict of interest.

## Abbreviations

The following abbreviations are used in this manuscript:

| | |
|---|---|
| HIV | Human immunodeficiency virus |
| AIDS | Acquired immune deficiency syndrome |
| MDPI | Multidisciplinary Digital Publishing Institute |
| DOAJ | Directory of open access journals |

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
