# Peer review of "Mathematical Model Describing HIV Infection with Time-Delayed CD4 T-Cell Activation"

_processes, doi:10.3390/pr8070782_

Round 1

Reviewer 1 Report

Detailed comments for Authors are reported in the attached file "ReviewForAuthors.pdf".

Reviewer 2 Report

The paper entitled"Mathematical model describing HIV infection with time-delayed CD4 T-cell activation" is well written and the topic is very interesting. A mathematical model that describe the population dynamics of CD4 T cells in the human immune system, as well as viral HIV particles, is proposed. And the equilibria stability is performed and numerical simulations are carried out. Also, the model is modified with and without activation time. The investigation contribute to the comprehension of the effect that the activation time has on the infected cells in the virus infection dynamic. It is also verified numerically. Clearly, the goal of the paper is quite desirable, and the results obtained may be reasonable. In my view, the paper is acceptable for publication. 

Author Response

The reviewer checked the box: 'English language and style are fine/minor spell check required'

A general style review of the entire paper was carried out. Inaccuracies or typos were corrected throughout the document.

Round 2

Reviewer 1 Report

The current version of the paper is definitely improved: typos have been fixed, additional comments have been inserted and the major revisions have been accomplished. So, I recommend the publication of the paper in the journal “Processes” after some minor revisions.

Minor revisions

  1. Lines 218, 219, 220: please fix and simplify the sentence, since it sounds awkward. For instance “Several numerical solutions are shown and depicted by means of solid black lines that describe the state trajectories toward E0. In this scenario, the virus–free equilibrium presents a stable node behavior”. Please correct “an stable” into “a stable”.
  2. Line 221: please remove the point before “i.e.”.
  3. Line 234: please split the sentence as “…viral load per mm3. From there…”.
  4. Line 235: please correct “show” into “shown”.
  5. Line 235: maybe the sentence should be simplified. For instance you can say: “In the bottom panel, the phase portrait shows the stable spiral-type behavior of the solution (solid black line).”
  6. Equation between lines 309-310: the computed value of the parameter dependent ratio is actually the value of the threshold “a0, not the value of the parameter “a”. “a0” is a function of the model parameters while “a” is a model parameter that can be independently fixed over or under the threshold “a0”. So you have to write “a0 = …”, not “a = …”. Please fix this point, since it is evidently wrong.
  7. Lines 319, 320, 321: please split the sentence. For instance “…activation time. Certainly, the magnitude…”.
  8. Line 344: “The study … contributes”.
  9. Line 347: “a scenario”, not “an scenario”.
  10. Lines 357-361: Please modify the sentence since it is not clear what you mean. First of all you could split at the point “…modelling the HIV infection.”, and, from this point on, rewrite your sentence better explaining your observations.

Author Response

Here I do provide a point-by-point response to the reviewer:

1. Lines 218, 219, 220: The suggestion was accepted and the correction was made.

2. Line 221: The correction was made.

3. Line 234 (now 233): The suggestion was accepted and the correction was made.

4. Line 235 (now 234): The text changes according to the suggestion given in 5.

5. Line 235 (now 234): The suggestion was accepted and the correction was made.

6. Equation between lines 309-310: Indeed the calculated value is a0 not a, the correction was made.

7. Lines 319, 320, 321 (now 320). The suggestion was accepted and the correction was made

8. Line 344 (now 343): The correction was made.

9. Line 347 (now 346): The correction was made.

10. Lines 357-361 (now 358-362). We have rewritten the text to make ideas more clear.

On my behalf and on behalf of the other authors, I want to express my sincere thanks to the reviewer, since the contributions have been very valuable and have contributed significantly to improving the quality of the article.

Hernán Darío Toro-Zapata
